

The effect of a short oxygen exposure period on algal biomass degradation and
methane release from eutrophic and oligotrophic lake sediments
Sigrid van Grinsven[1*], Natsumi Maeda[1], Clemens Glombitza[2], Mark A. Lever[2],
Carsten J. Schubert[1,2]
[1]Department of Surface Waters – Research and Management, Swiss Federal
Institute of Aquatic Science and Technology (EAWAG), Seestrasse 79,
Kastanienbaum, 6047, Switzerland.
[2]Institute of Biogeochemistry and Pollutant Dynamics, Swiss Federal Institute of
Technology, Zurich (ETH Zurich), Universitätstrasse 16, Zurich, 8092, Switzerland.
*Current address: Geomicrobiology, Department of Geosciences, University of
Tübingen, Germany, sigrid.van-grinsven@geo.uni-tuebingen.de





**Abstract**
Algal blooms in lakes result in large amounts of labile carbon being transported down
the water column towards the sediments, often resulting in temporary water column
hypoxia. The algal biomass is deposited at the surface sediments, where it is degraded
by the microbial community. Negative effects of algal blooms and biomass depositions
are sometimes mitigated by pumping air or oxygen into the bottom waters of lakes.
The fate of the algal biomass, in terms of greenhouse gas release, is however often
unknown. We investigated methane emissions from sediments originating from both
a eutrophic and oligotrophic lake and tested the effect of additional algal C inputs.
Additionally, we investigated the effect of a pulse supply of oxygen, a mediating
measure that is currently being used in the investigated eutrophic lake. Our results
show a difference in the control experiments based on the state of eutrophication, but
the methane release from new algal biomass additions was the same, although the
process proceeded more rapidly in the eutrophic sediments. A 3-week pulse of oxygen
lowered the emitted methane from both types of sediments by 50%, not only reducing
the emissions of algal biomass additions, but also reducing methane emissions from
the experiments without fresh organic matter inputs. This effect was relatively long-
lasting: its effects were visible for several weeks after anoxic conditions were re-
established, making it a potentially interesting measure to lower methane emissions
over a longer period.



**Introduction**

Lakes are known to be significant contributors to global methane emissions, despite their relatively small surface area(Bastviken et al. 2004). Methane emissions are the result of the net outcome of two processes: the production of methane, called methanogenesis, and the consumption of methane, methanotrophy. Methane emissions from aquatic environments originate mostly from sediments. Organic matter is delivered from either internal lacustrine (autochthonous) or external, e.g. riverine and terrestrial (allochthonous) sources. Autochthonous primary production, e.g. by planktonic microalgae in the water column, captures $CO_2$ from the atmosphere to produce biomass. After cell death, this biomass sinks down the water column, and becomes part of the sediment. The decomposition of this biomass lowers dissolved oxygen concentrations in bottom water and surface sediments and frequently enhances sedimentary methane production (Fiskal et al. 2019; van Grinsven et al. 2022). Eutrophication, the increase in (mainly algal) primary production due to increased nutrient concentrations in lakes, has thus been shown to increase methane emissions from lakes (Beaulieu, DelSontro, and Downing 2019).

Most sublittoral lake sediments are anoxic from a depth of a few mm to cm below the sediment surface. This is due to the mainly diffusive transport of oxygen into sediment and the high rates of aerobic decomposition processes at the sediment surface. In the underlying anoxic sediment, organic matter breakdown is performed by a community of hydrolytic, fermentative, and respiring microorganisms. Generally, methanogenesis is expected to occur only after the depletion of other, more energy-rich anaerobic oxidants, such as nitrate, nitrite, metal-oxides and sulfate (Bastviken et al. 2004),



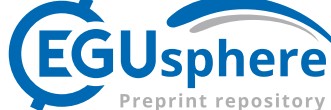

though strong overlaps in the distribution of methanogenesis with other anaerobic
respiration reactions have also been observed (Fiskal et al. 2019).

Generally, the anaerobic breakdown of organic matter follows three steps: first, the
large, complex organic compounds (e.g. macromolecules, polymers) are broken down
to their building blocks (e.g. monomers, oligomers, fatty acids) by extracellular
reactions (e.g. hydrolysis). Subsequently, these building blocks are taken up by
microbial cells and fermented to smaller chemical compounds, such as $H_2$, alcohols
(e.g. methanol, ethanol) and volatile fatty acids (VFAs; e.g., acetate, propionate,
butyrate, isovalerate, formate, and pyruvate). These smaller compounds can then
undergo a secondary fermentation step by syntrophic microorganisms, prior to
respiration, or be directly respired to $CO_2$ or methane using nitrate, nitrite, metal-
oxides, sulfate, or $CO_2$ as electron acceptors. Methanogens, which are respiring
organisms that gain energy through the production of methane, are mostly obligate
anaerobes, although their tolerance to oxygen is debated and may differ between
clades (Kato, Field, and Lettinga 1993; Zinder 1993; Kiener and Leisinger 1983).
Methanogenesis in sediments is believed to proceed mainly via three different
pathways: $CO_2$ reduction using H2 or formate as electron sources, acetoclastic
involving the disproportionation of acetate into $CO_2$ and methane, or methylotrophic
methanogenesis, which involves the conversion of methylated compounds, such as
methanol, methylamines and methyl sulfides to methane. With few exceptions, these
pathways are performed by distinct taxa of methanogens.

A significant fraction of methane is consumed by methanotrophy, the process of
methane consumption. Methanotrophy can occur both aerobically or anaerobically via



the reduction of various anaerobic electron acceptors. Methanotrophs are found both
within the archaeal and bacterial domain and include strict aerobes, facultative
anaerobes, and obligate anaerobes. Methanotrophic activity often peaks at the oxic-
anoxic interface in either the sediment or water column, presumably due to the high
energy yields of aerobic methanotrophy.

Past research  indicates that high organic matter inputs to lake sediments following
algal blooms result in increases in methane concentrations in the sediment (Schulz
and Conrad 1994). Both eutrophication and reduced water column mixing due to
increased thermal stratification as a result of global warming will likely increase the
frequency of algal blooms and contribute to more widespread bottom water anoxia  in
the future (Hou et al. 2022). While this promotes the deposition and burial of organic
carbon in lake sediments, it will also increase methane emissions by increasing
methanogenesis rates and lowering rates of aerobic methanotrophy. In addition to
future algal blooms, legacy effects, e.g. continued high rates of methane production
sustained by the decomposition of older organic carbon from past periods of
eutrophication may contribute to these elevated methane emissions. These increases
in methane emissions may be a lesser concern for oligotrophic sediments, which are
generally lower in organic carbon content and hence methane production rates.

In order to mediate the effects of current and past eutrophication, artificial aeration is
applied to lakes in Switzerland. This aeration reduces the detrimental ecological and
socioeconomic consequences of seasonal anoxia. In addition, artificial aeration may
lower methane emissions from lakes by promoting aerobic methanotrophy and
reducing methane production in deep water columns and surface sediments, though





the efficacy of artificial aeration in achieving lower methane emissions is not known.
Here, we experimentally test the impact of artificial aeration on methane emissions
under different trophic regimes by short, pulse-wise, oxygen supply to sediments from
oligotrophic Lake Lucerne and eutrophic Lake Baldegg (both Switzerland). Based on
slurry and whole-core incubations, we study the impact of oxygen pulses on methane
emissions from sediments with and without an initial spike of algal biomass to mimic
the situation shortly after an algal bloom. Samples were extracted for gas
concentration and isotope analysis, as well as VFA and microbial community analyses.
Our results show that oxygen exposure had effects lasting past the oxic period such
as that methane emission rates remained lower for up to 10 weeks. The methanogenic
and methanotrophic communities did not seem affected by the oxygen exposure.


**Methods**

Study sites
Two lakes with a different trophic state were sampled for various experiments. A map
showing the location of both lakes is shown in Fig. S1.
Lake Lucerne is located at the northern alpine front in Central Switzerland (47°N, 8°E,
434 m a.s.l). It has a surface area of 116 km$^2$, and is fed by four alpine rivers that
provide ±80% of the lakes total water supply (Schnellmann et al. 2002). It is
oligotrophic, with a maximum P-concentration within the past century of 1.7 μM (Bürgi
and Stadelmann 2002). Further details on the trophic history of both lakes can be
found in (Fiskal et al. 2019).



Lake Baldegg is in an area with intensive cattle and pig farming within Central
Switzerland. It has a surface area of 5.22 km². Eutrophication has been ongoing, with
a peak in the 1970s, reaching P concentrations of 15.4 μM before remediation
measures were put in place. It has had an anoxic hypolimnion for almost 100 years
until artificial aeration was started in 1982 (Gächter and Wehrli 1998).

Field sampling
Lake Lucerne sediment cores were taken in November 2020 and February 2021 at a
location near the village of Kastanienbaum (47.00085N, 8.33697E). Lake Baldegg
sediment cores were taken in June 2021 from the center of the lake (47.193071N,
8.265238E). Both lakes were sampled with a multicorer device containing 10 cm
diameter, transparent butyrate plastic core liners of 65 cm, which were never filled
more than 3/4rd. The average core length was 40 cm. All sediment cores were brought
into a climate room of 10°C within 2 hours after core collection and stored until further
processing. Bottom water temperatures are between 5 and 9°C, according to (Fiskal
et al. 2019).

**Slurry experiments in bottles**

| Oxygen regime | Biomass addition* | Lake | Headspace gas analysis | VFA | Microbial community |
|---|---|---|---|---|---|
| N$_2$ flushed at start | 0.1 g | Lucerne | + | + | + |
| N$_2$ flushed after 1 week | 0.1 g | Lucerne | + | + | + |
| N$_2$ flushed after 3 weeks | 0.1 g | Lucerne | + | + | + |
| N$_2$ flushed at start | - | Lucerne | + | + | + |
| N$_2$ flushed after 1 week | - | Lucerne | + | + | + |
| N$_2$ flushed after 3 weeks | - | Lucerne | + | + | + |
| N$_2$ flushed at start | 0.1 g | Baldegg | + | + | + |
| N$_2$ flushed after 3 weeks | 0.1 g | Baldegg | + | + | + |



| | | | | | |
|---|---|---|---|---|---|
| N₂ flushed at start | - | Baldegg | + | + | + |
| N₂ flushed after 3 weeks | - | Baldegg | + | + | + |

**Whole sediment core experiments**

| Oxygen regime | Biomass addition* | Lake | Headspace gas analysis | VFA | Microbial community |
|---|---|---|---|---|---|
| N₂ flushed at start | 0.3 g | Lucerne | + | - | - |
| No N₂ flushing, air headspace | 0.3 g | Lucerne | + | - | - |
| N₂ flushed at start | - | Lucerne | + | - | - |
| No N₂ flushing, air headspace | - | Lucerne | + | - | - |

\* freeze-dried Spirulina algae (slurries) or 1:1 mixture of freeze-dried Spirulina + Chlorella algae (whole cores)


*Table 1. Overview of experiments. Further details are provided in Table 2 and the*
*Methods section.*

Experimental setups
The experimental setups were designed to mimic oxygen intrusion from overlying oxic
water. Two different experimental setups were used: slurry experiments (Lakes
Baldegg and Lucerne) and whole core experiments (Lake Lucerne only). An overview
of the experiments and analyses is presented in Table 1. Oxygen concentrations were
followed and are shown in Fig. S2 and S3.

Slurry incubation experiments
The oxygen penetration depth of aquatic sediments generally does not exceed 1 cm
(Fiskal et al. 2019; Horppila et al. 2015), with outliers in very organic material poor
(marine) sediments up to 6 cm(Cai and Sayles 1996). It is highly unlikely that
sediments deeper than 5 cm will experience oxygen intrusion. Therefore, we choose



a setup which does not expose the sediments deeper than 5 cm to oxygen, as this
would inhibit methanogenesis in a way that is highly unlikely to appear in lake
sediments.
To ensure this, sediment cores were separated in a surface part (0 – 5 cm depth) and
a deep part (5 – 15 cm depth). Lake bottom water was used to dilute sediment material
1:1, to create slurries. For the experiments that started completely anoxically directly
from the start, both the surface sediment and deeper sediment were added at the
same time and flushed with $N_2$. For the experiments that started under oxic conditions
(see Table 1), only surface sediments were placed into the incubation vials. The
deeper sediments were added upon $O_2$ removal, which was after either 1 or 3 weeks.
An overview is provided in Table 2. The oxygen concentrations inside a subset of
incubation bottles was followed to ensure the oxic and anoxic periods were indeed
established as aimed for (Fig. S2).

| Treatment | Sediment provided at start | Sediment added after 1 week | Sediment added after 3 weeks | Final contents |
|---|---|---|---|---|
| N2 flushed at start | 0 – 15 cm | - | - | 0 – 15 cm |
| N2 flushed after 1 week | 0 – 5 cm | 5 – 15 cm | - | 0 – 15 cm |
| N2 flushed after 3 weeks | 0 – 5 cm | - | 5 – 15 cm | 0 – 15 cm |


*Table 2. Overview of sediment additions within different experimental treatments of*
*the slurry experiment.*

Slurry experiments were performed in triplicate in 0.5L Schott bottles, filled with a final
volume of 258 ml sediment slurry. Each bottle was closed with an adapted stopper,
containing a three-way-stopcock that could be opened to allow throughflow of
sediment slurry and gas without exchange with the air. Freeze-dried algal biomass
(0.1 g of freeze-dried Chlorella; DietFoods CH) was added to a subset of the bottles.





Incubations proceeded at 10°C in the dark. The oxygen concentration was measured
daily to bi-weekly in 8 out of the 12 oxic bottles, using optical oxygen sensor spots
(Pyroscience, UK).

After 1 week or 3 weeks, the oxic bottles were flushed with $N_2$ for 15 minutes to remove
oxygen, after which they received additional anoxic sediments, according to Table 2.
The resulting oxygen trends are shown in Fig. S2. Sediments were added via the
sampling ports in the stoppers.

At each gas sampling time point, headspace gas was sampled via the sampling port
in the bottle caps. Prior to gas extraction, 10 ml of $N_2$ or air was pushed into the bottle.
Immediately after, 10 ml of headspace gas was sampled and stored in $N_2$ flushed 70
ml serum bottles. At each VFA (volatile fatty acids) and DNA sampling time point, 1.5
ml overlying water and 1.5 ml mixed slurry were sampled as described below. For the
VFA samples, samples without particulates were required. To achieve this, the bottles
were carefully tipped over, so the natural layering of sediment at the bottom and water
at the top remained. The water was then sampled via the sampling port. For the DNA
sample, the bottle was briefly shaken and then held upside down to take a mixed water
+ sediment DNA sample. Both VFA and DNA samples were put directly on ice, and
after the sampling series was finished, moved to -20°C (VFA) or -80°C (DNA) freezers.

Whole core incubation experiments
In contrast to the slurry experiments, the cores for the whole core incubation
experiments were not disturbed or opened. Whole core experiments were only
performed with Lake Lucerne sediments for practical reasons, as shown in Table 1.



The core retrieval resulted in cores that were filled with on average 40 cm of sediment
and 25 cm of overlying water. To allow for headspace gas extraction over the
experiment duration, 12 cm of overlying water was carefully removed without
disturbing the sediment surface, leaving on average 13 cm of overlying water above
the sediment-water interface of each core. Whole core experiments were performed
in quadruplicate, an overview of the treatments is provided in Table 1.
To set up the treatments, freeze-dried algal biomass was added to selected cores ca.
18 hours after sampling (0.3 g per core, 1:1 mixture of freeze-dried Chlorella and
Spirulina; DietFoods, CH), corresponding to the same amount of algal biomass per
sediment mass as in the slurry incubations, and following earlier studies(Hiltunen,
Nykänen, and Syväranta 2021; Dai et al. 2005). The algal biomass was carefully
deposited on the sediment surface using a pipet, without disturbing the sediment-water
interface. The headspace and overlying water of the cores for the anoxic incubations
were flushed for at least 10 minutes with $N_2$.
Adjusted rubber stoppers with sampling ports were used to seal the cores on the top,
the bottoms were sealed off with rubber stoppers plus plastic caps. All cores were then
placed at 10°C in the dark. One core of each oxic treatment contained an oxygen
sensor spot (Pyroscience, UK), glued to the inner wall of the core liner.
Oxygen concentrations were measured at the start of the experiment and at irregular
intervals over the course of the experiment (shown in Fig. S3) and showed oxic
conditions were indeed retained over the full course of the oxic experiment, as was
aimed for.

Over the course of the experiment, gas samples for $CH_4$, $CO_2$ and $N_2O$ analysis were
taken via the sampling ports. 10 ml of the gas headspace was extracted and placed





into 70 ml $N_2$-flushed serum bottles with butyl stoppers. After the gas sampling, 10-15
ml of $N_2$ gas (anoxic cores) or $N_2$ or air (oxic cores) was added via the sampling ports,
to equilibrate the internal and external pressure and limit the risk of leakage or
contamination. The gas pressure prior to sampling was determined with a pressure
meter and noted for each timepoint (not shown). One core was discarded due to water
leakage during the experiment.

Gas concentration and stable isotope analysis
Gas samples were analyzed for the concentration of $CH_4$ and $CO_2$ by gas
chromatography (GC; Agilent 6890N, Agilent Technologies) using a Carboxen 1010
column (30 m x 0.53 mm, Supelco), a flame ionization detector and an auto-sampler
(Valco Instruments Co. Inc.) for both the slurry and whole core experiments. Isotopic
ratios of methane 13C/12C (presented in the standard δ13C-notation relative to the
Vienna Pee Dee Belemnite (VPDB) reference) were measured in selected headspace
samples by isotope ratio mass spectrometry (IRMS; GV Instruments, Isoprime). To
purify, concentrate and combust the $CH_4$ to $CO_2$, injected samples were passed
through a trace gas unit (T/GAS PRECON, Micromass UK Ldt).

VFA analysis
Volatile fatty acids (VFA) analysis was only performed on samples from the slurry
experiment. Samples were filtered through pre-cleaned syringe filters Acrodisc™, 0.2
μm PES membrane, Supor™) and analyzed by two dimensional ion chromatography
(2D IC) at ETHZ according to the method described in (Glombitza et al. 2014) with
some modifications. The instrument used was a Dinonex™ ICS6000 (Thermo Fisher
Scientific) equipped with two 2.5-mm columns (AS24 for the first dimension and



AC11HC for the second dimension). The Retention time window on the first IC
dimension to collect the bulk VFAs for injection onto the second IC column was set to
3 min – 6.5 min to account for the low salinity of the freshwater samples compared to
the original method, as described in (Schaedler et al. 2018; Vuillemin et al. 2023).
Likewise, the VFA standards for quantification (mixed standards of formate, acetate,
propionate, butyrate, valerate, isovalerate and pyruvate at 1, 5, 10, 50 and 100 µmol
$L^{-1}$) were prepared in Milli-Q® water instead of IAPSO seawater as described in the
original method. Quantification was done using the conductivity detector signal of the
second IC dimension.

Microbial community analysis
Microbial community analysis was only performed on samples from the slurry
experiment. Each DNA sampled was stored at -80°C until processing. DNA was
extracted using the Qiagen Powersoil DNeasy kit without adaptations. The DNA
concentration of all extracts was measured on a Nanodrop device (Thermo Scientific).
When the concentration was below 2 ng/µl, an additional extraction was performed,
and samples were pooled. DNA extracts from all experiments were combined in two
lanes, including extraction blanks, and send for 16S rRNA NovaSeq PE250
sequencing (30K tags per sample) to Novogene UK, using the general 16S rRNA
archaeal and bacteria primer pair 515F and 806R, targeting the V4 region (Caporaso
et al. 2012). Quality control and species annotation were performed using the standard
Novogene     pipelines     (https://www.novogene.com/eu-en/services/research-
services/metagenome-sequencing/16s-18s-its-amplicon-metagenomic-sequencing/).
Raw sequencing data is deposited in the public repository [available upon publication,
or on reviewers' request].





**Results**

*Slurry experiments oligotrophic Lake Lucerne*

All oligotrophic slurry experiments that received algal biomass emitted significant quantities of methane. The control experiment, without additional carbon source, only emitted methane in the permanently anoxic setup (ca. 9 μmol per week). The control slurries in which the top 5 cm was initially exposed to oxygen did not emit methane, also not after anoxic conditions were established and the deeper anoxic sediments were added (Fig. 1; Fig. 2). All given concentrations and concentration increases are given in μM per liter headspace volume.

The addition of algal biomass to the oligotrophic slurries increased the methane emission almost 25-fold under permanently anoxic conditions (to 161 μM per week), but only 17-fold and 14-fold under the 1-week and 3-week oxic start conditions (118 μM and 93 μM per week), respectively (Fig. 1, Fig. 2).

Methane emissions started directly in the first week after the start of the anoxic experiments (Fig. 2). After oxygen removal from the oxic start slurries, thus re-establishing anoxic conditions and adding deeper sediments, methane emission also started immediately. Net methane emission continued until week 13 in the anoxic oligotrophic slurries, and two weeks longer in both types of the oxic slurries, despite the 2 weeks difference between the start of methane emission in these two oxic incubations (Fig. 2). The methane concentration in the oligotrophic slurries plateaued and remained constant between weeks 13 or 15 to week 28, at a concentration of 1900 (anoxic), 1500 (1-week oxic start) or 1200 (3-weeks oxic start) μmol per L headspace.



*Slurry experiments eutrophic Lake Baldegg*
Methane emission was observed in all eutrophic slurry setups, both the control and
algal biomass addition setups, after establishment of anoxic conditions and the
addition of deeper sediments. Under oxic conditions, and with only the top 5 cm, no
methane emission was observed. However, similar to the oligotrophic setup, the onset
of methane emission happened directly after establishment of anoxic conditions and
the addition of the deeper sediments.
The anoxic background methane emission in the eutrophic control slurries exceeded
those of the oligotrophic control slurries over 12-fold (85 versus 6.9 μM per week,
respectively, Fig. 1; Table S1).

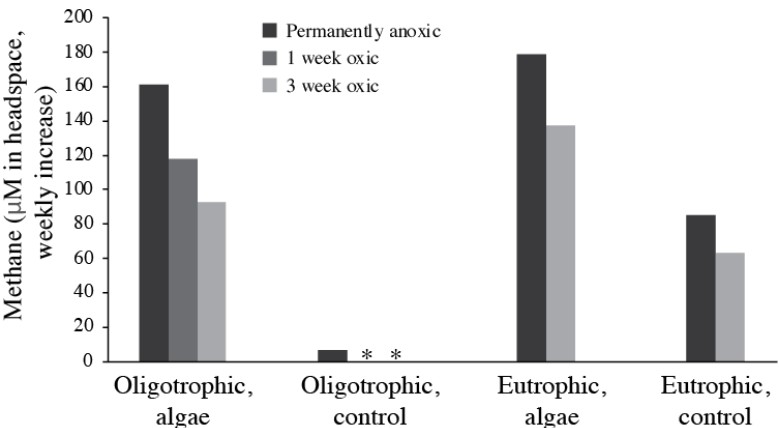



Fig. 1. Weekly increase in headspace methane concentration in the oligotrophic and
eutrophic slurry experiments, respectively, as derived from the linear phase of the methane
concentration plots (Fig. 2). * no increase in concentration detected.

The methane concentration in the eutrophic slurries did not plateau, although two
phases could be identified in the algae-fed slurries: A phase of rapid increase in





headspace $CH_4$ was observed from $t_0$ until week 7 in the anoxic algae-fed slurries, and
from week 3 to week 9 in the 3-week oxic start algae-fed slurries (Fig. 2). After this
initial phase, the methane emission rate stabilized at a similar rate as was observed
in the control experiments, as can be observed by the parallel lines in the graph of Fig.

342    2.


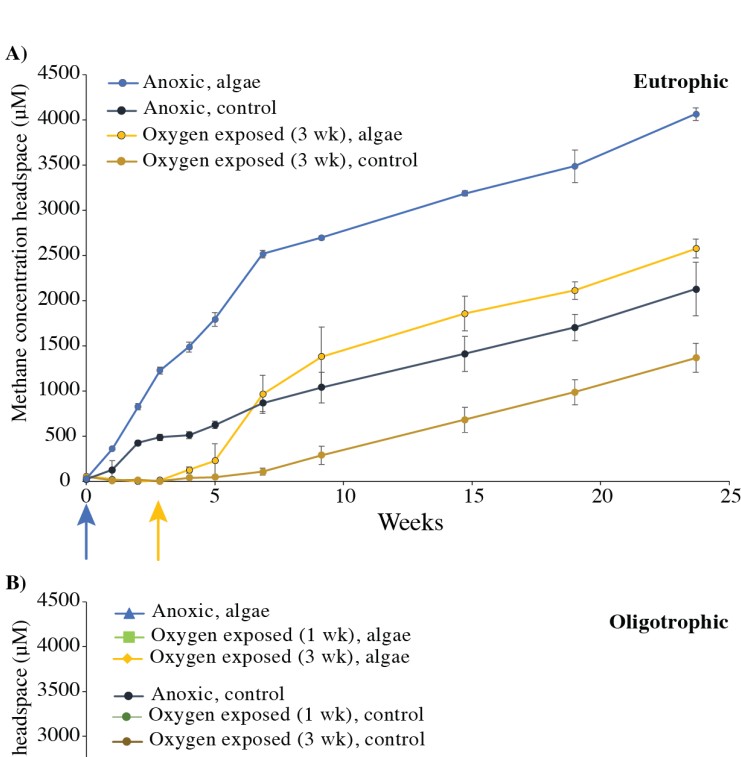





**Fig. 2**. Methane concentration in the headspace of the slurry experiments with A) eutrophic
and B) oligotrophic sediments. Arrows indicate the moment of oxygen removal, by flushing
with $N_2$, and the addition of anoxic sediments (see methods).

Even though the addition of algal biomass did increase the methane emission from
the eutrophic slurries, it was partly diminished when oxygen was present at the start
of the experiment. The background methane emission under anoxic control conditions
was similar to that of the oxic conditions with additional algal biomass, resulting in a
total amount of methane emitted of 2600 and 2100 $\mu$M for the oxic with algae and
anoxic control, respectively, corresponding to an increase of only 20% in emitted
methane (Table S1; Fig. 2). The methane emission rate was initially higher in the oxic
incubations with algae, but because the phase of high emission was of a short duration
(6 weeks, from week 3 to week 9), the total emission did not strongly exceed the total
anoxic control emission. The high emission phase in the anoxic algae experiment was
also short ($t_0$ until week 7), but due to the high weekly emission rate of 180 $\mu$mol, the
total amount of methane produced after 24 weeks was twice as high as the methane
emission in the anoxic control experiment, and 1.5 times higher than in the oxic algae
experiment (Fig. 2; Table S1).

Both the oligotrophic and eutrophic incubation experiments received equal amounts
of algal biomass. The methane emission rate in the oligotrophic anoxic experiments
increased from 6.9 to 161 $\mu$M per week due to the algae addition, whereas the
eutrophic anoxic rate increased from 85 to 179 $\mu$M per week, showing a much larger
increase in the oligotrophic experiments. The same holds for the 3-weeks oxic
experiments, which increased in weekly rate from 0 (control) to 93 (with algae) in the



oligotrophic experiments and from 63 to 137 in the eutrophic experiments,
respectively. Due to the shorter duration of this high-rate methane emission the total
methane produced as a result of the algal biomass was, however, similar between the
oligotrophic and eutrophic sediments (Fig. 2).

*Intact core experiments oligotrophic Lake Lucerne*
Experiments with whole sediment cores, rather than sediment slurries, showed a
similar effect of oxygen exposure and algal biomass additions as the oligotrophic slurry
experiments. Although the variation within the experiments with whole cores was much
larger than in the more controlled slurry experiments, still a significant effect of oxygen
exposure, and of algal biomass addition, was observed (Fig. 3).

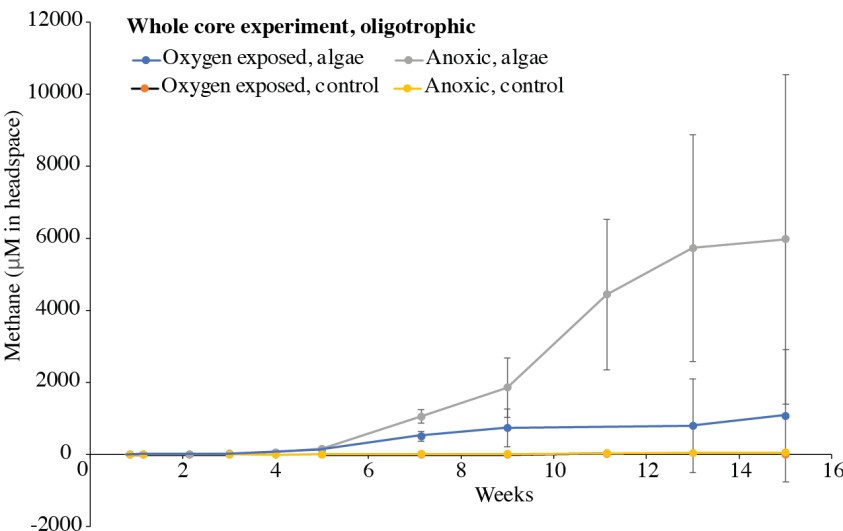



**Fig. 3.** Methane concentration in the headspace of the whole core experiment under
oxygen exposed (air headspace above overlying water) or anoxic ($N_2$ headspace)
conditions, with and without the addition of algal biomass, respectively. A cut-out of





the lower values that highlights the methane concentrations at the start of the
experiments is available in Fig. S4. The oxygen exposed control line is hidden from
view behind the anoxic control line in this graph.

**Microbial community in slurry experiments**

*Effect of algal biomass on microbial community*
The phyla of the Proteobacteria and the Bacteriodota were abundant in both the
eutrophic and oligotrophic sediment incubations. Other phyla that were found among
the 10 most abundant phyla in both setups were the Verrucomicrobiota, Chloroflexi,
Acidobacteriota, Desulfobacterota and Planctomycetota.
The microbial community was similar in the permanently anoxic and initially oxic
treatments, in both the oligotrophic and eutrophic sediment incubations. The addition
of the algal biomass influenced the microbial community composition. In the
oligotrophic incubations, the relative abundance of the Proteobacteria was higher in
all incubations with algae, both with and without oxygen exposure. In the eutrophic
experiments, the Proteobacteria abundance was actually lower in incubations with
algal biomass than without, both oxygen exposed and permanently anoxic. The
abundance of the Bacteroidota was higher in incubations with algal biomass than the
control, in all treatments in both setups. The Nitrospirota had a high abundance in
the oligotrophic sediments (11 – 14% in control setups), that was lowered under
conditions with algal biomass (8 – 10%, Fig. 4). In the eutrophic sediments, the
relative abundance of Nitrospirota was <1%. The abundance of the Acidobacteriota
was the same in oligotrophic sediments with and without algal biomass (5 – 7%). In



the eutrophic sediments, the relative abundance in the control incubations was lower
(3%) than in the algal addition incubations (7%).

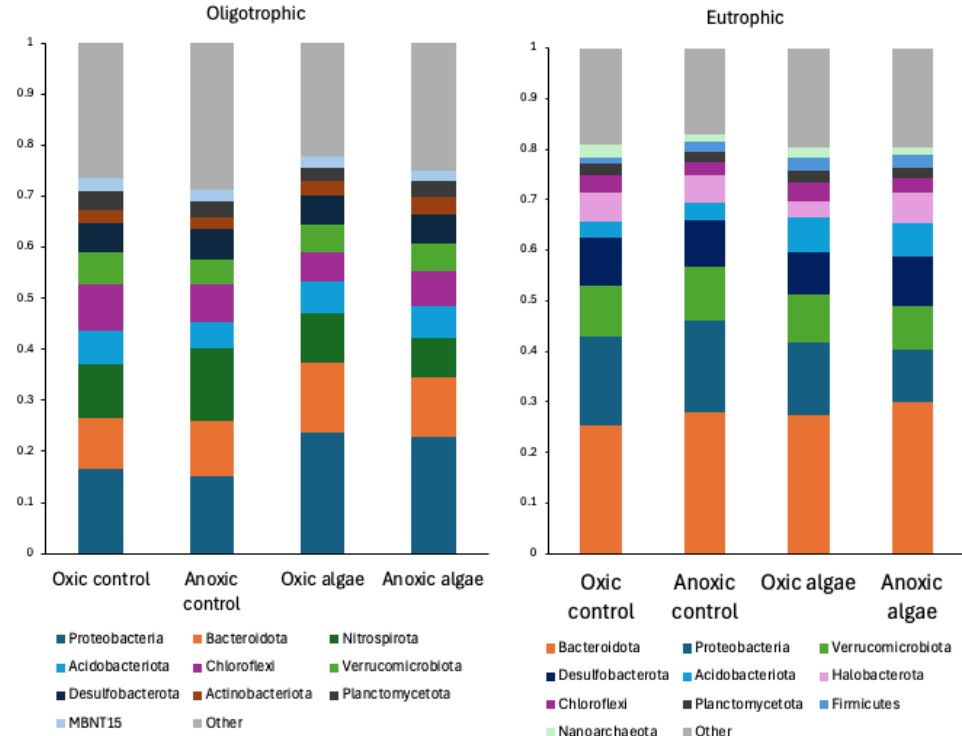


**Fig. 4.** Microbial phyla in the oligotrophic (A) and eutrophic (B) sediment incubations,

after 15 weeks and 9 weeks, respectively. The 10 most abundant phyla, as detected
by 16S rRNA sequencing, are shown. The timepoints correspond to the start of the
stationary methane release phase in both setups, as shown in Fig. 2.

*Methanogenic/methanotrophic clades*
Surprisingly, nor the oxygen exposure or the algal biomass additions had a significant
effect on the relative abundance of methanogenic and methanotrophic clades in the
microbial   community   (Fig.   S5).   There   was   a   profound   difference   between   the
oligotrophic and eutrophic sediments, but only for the methanogen relative abundance.



In the oligotrophic incubations, the relative abundance of operational taxonomic units
(OTUs) assigned to methanogenic orders was between 0.5 and 3% of the total 16S
rRNA detected sequences (Fig. S5) at all tested timepoints between 0 and 27 weeks.
In the eutrophic experiments, the relative abundance of OTUs assigned to
methanogenic orders was mostly between 4 and 7% of the total 16S rRNA detected
sequences in all selected timepoints between 0 and 15 weeks (Fig. S5).

The methanogenic community was dominated by OTUs assigned to the order
Methanomicrobiales (Fig. S6). The contribution of Methanomicrobiales to the total 16S
rRNA sequences assigned to methanogenic orders was, however, higher in the
eutrophic (average of 75%) than in the oligotrophic experiments (average of 65%).
Methanomassiliicoccales made up a larger fraction in the oligotrophic experiments
(average of 29%, versus 12% in eutrophic experiments). Methanosarcinales were
relatively more abundant in the eutrophic experiments (Fig. S6). In the oligotrophic
experiments, no patterns were observed over time, nor differences between
treatments. In the eutrophic experiments, Methanomicrobiales dominated both the
oxic and anoxic experiments. However, in the permanently anoxic incubations, more
OTUs are assigned to Methanosarcinales, whereas in the temporarily oxic
incubations, more OTUs are assigned to the order Methanomassiliicoccales.

In both the oligotrophic and eutrophic incubations, the relative abundance of
methanotrophs belonging to the order Methylococcales was around 1 – 3% for all
treatments (Fig. 5). The majority of OTUs within the Methylococcales order were
assigned to the genus *Crenothrix* in all eutrophic treatments (20 – 99% of
Methylococcales reads), followed by the genus *Methylobacter* (0.9 – 30.6% of





*Methylococcales* reads, Fig. 5). However, recent research on the SILVA annotation
within the Methylococcales order has shown that a differentiation between *Crenothrix*
and certain groups of *Methylococcaceae* cannot be supported, and the weight of the
differentiation between these two groups is thus limited, currently (van Grinsven et al.
2022). The methanotrophic community in the oligotrophic experiments was less
dominated by sequences assigned to "*Crenothrix*", and rather than a higher
abundance of *Methylobacter* assigned OTUs (Fig. 5) and a higher abundance of
genera that were marginal in the eutrophic incubations, such as *Methyloparacoccus*.
At specific time points, the genus *Methylomonas* showed particularly high peaks in its
relative abundance (up to 74% of Methylococcales reads) in both the eutrophic and
oligotrophic incubations (Fig. 5). Overall, the methanotrophic communities looked
relatively similar and no trends could be established over time, nor differences
between the oxygen or algae treatments (Fig. 5, Fig. S7).





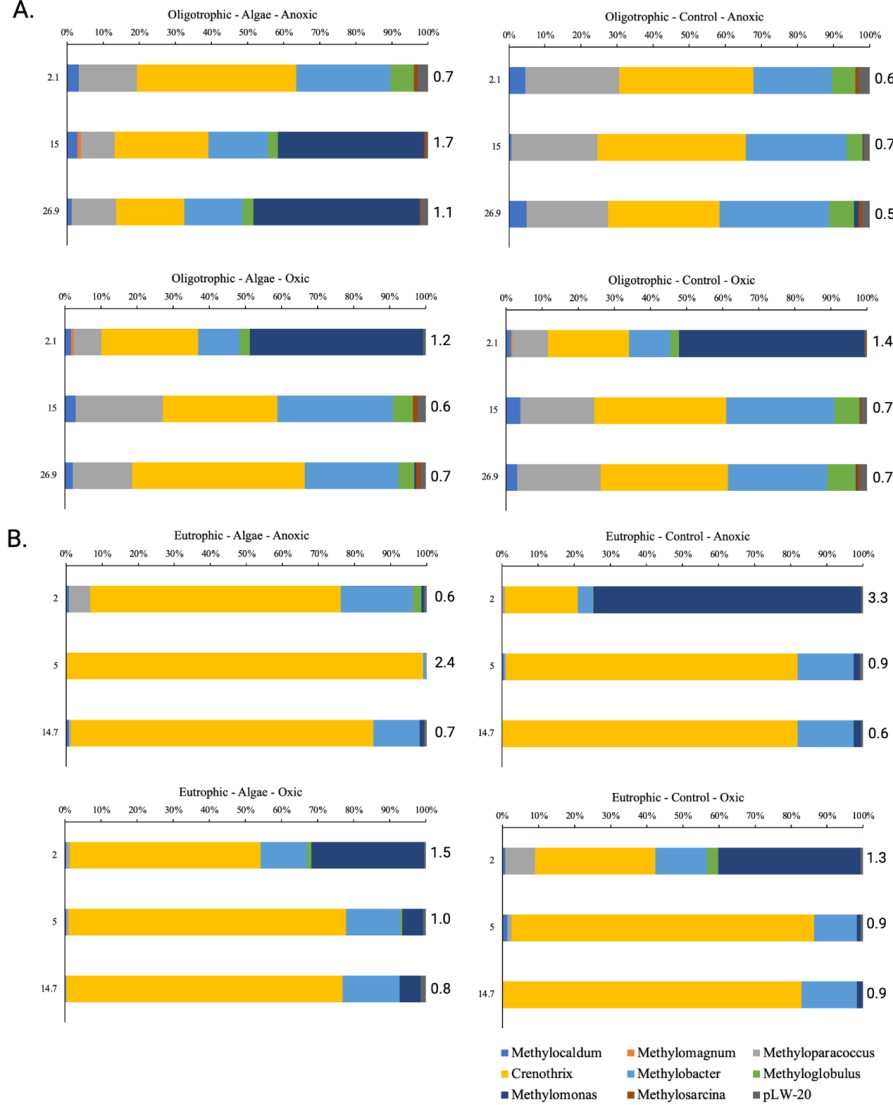

**Fig. 5.** Relative abundance of OTUs assigned to methanotrophic genera in the

oligotrophic (A) and eutrophic (B) incubation experiments. The bar plots show the

abundance of specific genera relative to the total methanotroph abundance, whereas

the number behind each bar indicates the abundance of methanotrophs relative to





the total microbial community (in % of 16S rRNA reads of each sample). The y-axis
shows the time in weeks since the start of the experiment.

**471    Volatile fatty acids in oligotrophic incubation experiments**


To compare the release of different volatile fatty acids (VFAs) after the algal biomass
additions, VFA concentrations were traced in the oligotrophic slurry experiments
during the first 8 weeks, as shown in Fig. 6. VFA concentrations were highest in anoxic
incubations, and were significantly lowered by temporary oxygen exposure. The
addition of algal biomass led to a strong increase in VFA concentrations (5 - 500 fold
increase) compared to control incubations. One of the anoxic control incubations had
20-fold higher VFA concentrations than the other two bottles of this treatment, resulting
in the large error bars (Fig. 6A).
Acetate was the dominant VFA, with concentrations 10-100x higher than the other
VFAs (Fig. S8), and the key VFA to differ between oxic and anoxic treatments. The
concentrations of formate and pyruvate were not significantly affected by the oxygen
exposure.



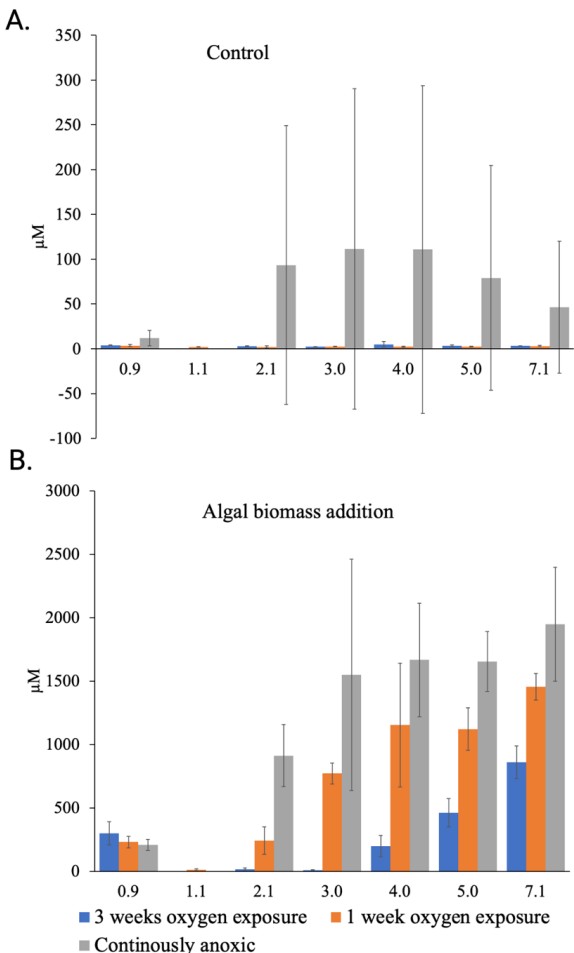

**Fig. 6.** Total volatile fatty acid concentrations during the first 8 weeks of the oligotrophic control (A) and algal biomass addition (B) experiment. The x-axis indicates weeks since the start of the experiment. Note the different y-axis for A and B. Each bar represents the average of triplicate samples at each timepoint. No samples were taken at 1.1 weeks (8 days) of the anoxic and 3 weeks-exposure experiments.

**Carbon isotopes of methane in slurry incubations**



The stable isotope profile of the headspace methane was determined only in the
stable, post-algal biomass degradation phase in the slurry experiments, when
methane emission rates no longer increased over time. This was at 17 weeks in the
oligotrophic, and at 11 weeks in the eutrophic experiments. The stable isotope ratio is
given in the standard δ13C-notation, relative to the Vienna Pee Dee Belemnite (VPDB)
reference. When comparing the treatments, we see more negative $\delta^{13}CH_4$ values in
the algal biomass experiments in both the eutrophic and oligotrophic slurry
experiments, compared to the controls. A significant difference between the oxygen
exposed and anoxic treatments was only visible in the oligotrophic sediments.


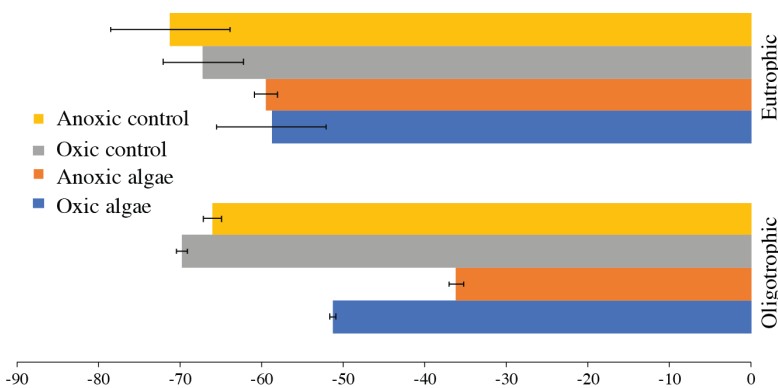

*Fig. 7. $\delta^{13}CH_4$ values of headspace methane, after 11 weeks (eutrophic sediments)*
*or 17 weeks (oligotrophic) of slurry incubations.*


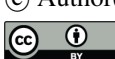


## Discussion

*Trophic state and legacy effects on methane emissions and methane-cycling communities*

The two investigated lakes differ in their current and historical trophic status. Lake Lucerne is currently oligotrophic and has a history of low phosphate inputs. Lake Baldegg is eutrophic and has been receiving high phosphate loading in the past, resulting in water column anoxia and algal blooms in the period of 1910 – 1985(Fiskal et al. 2019). Our results show that the eutrophication state of the lake affects the methane emissions throughout the entire incubation period, both with and without fresh organic matter inputs. The importance of legacy effects on biogeochemical processes and communities in these lakes has been shown in earlier studies as well (Fiskal et al. 2019; Han et al. 2020) and was also shown for other lakes along a trophic gradient (Zhou et al. 2024). The methane emission was 12 times higher in control experiments with eutrophic sediments than oligotrophic sediments, though no fresh material input was delivered over 160 days (Fig. 1, Table S1). Contrastingly to their historic and current carbon inputs, the TOC concentrations in sediments of both lakes are comparable, according to a recent study at the same sampling locations as used in this study (Fiskal et al. 2019).

Although these legacy effects are clearly visible in the methane emissions of the control setups, the response to the input of new algal material was similar in magnitude in both oligotrophic and eutrophic sediments (Fig. 1; Fig. 2). The addition of easily degradable carbon compounds to environmental samples can spark a priming effect, in which carbon stocks in the original sample are degraded more rapidly upon addition



of additional fresh material (Y. Wang et al. 2021; Guenet et al. 2010). Although we
cannot separate the contribution of algal carbon and sedimentary carbon to the
emitted $CH_4$, the equal emission responses from both lakes to the addition of fresh
organic matter suggest that the OM degradation response was also similar, and likely
primarily driven by degradation of the algal biomass rather than by older sedimentary
organic matter. A study with a similar setup regarding the addition of algal biomass
also showed a direct response in methane production rates during the first 60 days,
after which methane production rates stabilized (T. Wang et al. 2023).

Previous research has shown higher abundances and diversities of methanogens and
methanotrophs in eutrophic than in oligotrophic sediments(Yang et al. 2019), although
other studies indicate that the effect of sediment depth is of stronger effect(Yang et al.
2017) than the trophic state of the lake. The continuous high methane emission in our
eutrophic sediments (weekly increase of 85 $\mu$M in control experiments), in contrast to
the low methane emission in the oligotrophic sediments (weekly increase of 7 $\mu$M in
control experiments), also suggests that a more abundant and more active
methanogenic microbial community may exist in the eutrophic sediments. The
methanogenic community was indeed higher in its relative abundance (3 – 7%) in the
eutrophic experiments than in the oligotrophic sediments (0.5 – 4%).

*Effect of algal biomass and oxygen exposure on substrate availability to the microbial*
*community*
The conversion of algal biomass to gaseous methane emissions requires an initial
step of carbon degradation by fermenters, and a second step in which the reaction
products are converted into methane. The reaction products can consist of various



organic molecules, of which part can be used by methanogens directly, but others
need to be degraded further to become a suitable substrate for methane production.
Here, we found that the concentration of organic compounds indeed increased
strongly after the addition of algal biomass to the slurries, up to 500-fold (Fig. 6). This
was similar to a study by Schwarz et al. (Schwarz, Eckert, and Conrad 2008), who
found increased acetate and propionate concentrations in lake sediment incubations
with algal additions. A study by (Zhou et al. 2024) showed algal deposition on top of
the surface sediments led to a distinct increase in TOC in the top 8 cm of the sediment
cores, also without active mixing. The same two compounds as found in the Schwarz
et al. 2008 study, acetate and propionate, were also the major compounds detected
in our experiments. Surprisingly, these compounds were also produced (and build up)
under oxic conditions (200 $\mu$M acetate and 13 $\mu$M propionate under oxic conditions at
day 6, Fig. S8). However, the production of both acetate and propionate did not reach
the same values as the concentrations reached in the continuously anoxic incubations,
and a clear difference in the VFA buildup was also visible between 1 or 3 weeks
oxygen exposure treatments. The short exposure to oxic condition did lower the
acetate and propionate buildup, but did not diminish it. A recent study by (Kallistova et
al. 2023) showed that acetate additions strongly enhanced methane production from
surface sediments, showing it had an active function as methane precursor and higher
concentrations of acetate are likely directly correlated to higher methane emissions
from the sediments into the water column. In our experiments, substrates for acetate-
consuming methanogens were present in both the oxygen-exposed and permanently
anoxic experiments, but the concentrations were significantly lowered by oxygen
exposure at the start of the experiment. This corresponds to the methane production
in each of these treatments (Fig. 2). The methanogenic community did not show



similar patterns, suggesting that the substrate concentrations rather than the microbial
presence determines and predicts the methane emission rates in lake sediments.

*Microbial community*
A recent study by (Yang et al. 2021) followed the succession of a sedimentary
microbial community during an algal bloom, and found that both the archaeal and
bacterial community transitioned, taking part in biomass degradation steps that
changed over the time since the start of the algal bloom. (Schwarz, Eckert, and
Conrad 2008) noted that in their experiments with algal biomass additions, the
Deltaproteobacteria and Clostridiales increased immediately, and the Bacteroidetes
after 6 days. Methanogens, specifically acetate-using methanogens of the type
Methanosaetaceae increased in abundance after 6 days(Schwarz, Eckert, and
Conrad 2008). In our experiments, we noticed a similar increase in the
Proteobacterial relative abundance in the oligotrophic incubations due to algal
biomass additions, but a decrease in the eutrophic sediments (Fig. 4). The
Bacteroidota also increased in abundance in the oligotrophic incubations only. We
did not see an increase in the relative abundance in methanogenic clades in
response to the algal biomass addition, despite the much higher emission rates. A
similar effect was observed in experiments with algal biomass additions by (T. Wang
et al. 2023), who also saw large effects on the methane emission, but no increase in
methanogen copy numbers.
The effect of the oxygen exposure on the microbial community composition was
limited, both on the total prokaryotic community and on the specific methane-cycling
community. The methanogenic archaea are predominantly present in deeper
sediment layers (> 5 cm depth, as shown for these lakes by (Meier et al. 2024). They



will therefore likely not be affected by oxygen supply to lake bottom waters, which
was the process that was mimicked here. In our experimental setup, only sediments
of 0 – 5 cm depth were oxygen exposed, and 5 – 15 cm depth sediments were
added after oxygen removal. Therefore, the methanogenic community was
predominantly affected via the availability of substrates, and not due to direct oxygen
toxicity. Methanotrophs were presented throughout the sediment in these lakes (van
Grinsven et al. 2022). The methanotrophic bacteria found in our experiments are all
know as aerobic methanotrophs. However, these methanotrophs have been found in
anoxic environments, in these lakes and others, more often (van Grinsven et al.
2022). Although the methanotrophic community was diverse, especially in the
oligotrophic sediments, and showed changes in structure over time, these did not
seem related to oxygen exposure (Fig. 5).

*Oxygen exposure decreases methane emissions*
The effect of oxygen penetration depth on methane emission from lake sediments is
well established. However, these studies generally address long term stable oxygen
conditions ((Sobek et al. 2009; Huttunen et al. 2006). Here, we look at short oxygen
pulses, as a potential mediative measure for lakes with anoxic bottom water. The
presence of oxygen for a short, 3-week period at the start of the incubation had major
implications for methane emissions over the course of the entire experiments. The
total release of methane was significantly lower in the treatments that had experienced
an oxic period (Fig. 1; Fig. 2, Table S1). Most likely, part of the algal biomass was
converted to $CO_2$ and/or biomass during the oxic period and was therefore not directly
available for methanogenesis anymore. This is supported by the peak in $CO_2$
emissions that was observed during the oxic period of the experiments (Fig. S9; S10).





However, due to difficulties in translating headspace $CO_2$ concentrations to dissolved
$CO_2$, it is not possible to make a carbon mass balance, to see how much is indeed
released as $CO_2$. Part of the produced $CO_2$ will again be converted prior to release to
the headspace, leading to underestimates that cannot be sufficiently quantified. The
bubbling with $N_2$ to remove oxygen, that occurred at different timepoints in the different
experiments, removed $CO_2$ and may therefore has changed the pH in the system. pH
was not measured. Given the immediate production of $CO_2$ after bubbling (Fig. S9,
S10), we however assume that a $[CO_2]$ close to natural conditions was rapidly
established following $N_2$ bubbling.
As $CO_2$ has a much lower warming potential per mole than methane (approximately
28 times lower on a hundred year basis, (Forster et al. 2021) ) the release of $CO_2$ is
strongly preferred over that of methane in light of global warming. Besides $CO_2$, part
of the carbon may have been converted to microbial biomass during the oxic period,
and is stored as such in the sediments. (Sobek et al. 2009) published a weak linear
relationship between the diffusive methane flux from lake sediments, and the oxygen
penetration depth at those locations. A direct comparison with this study is, however,
difficult to make, as there are likely other factors involved that affect both the oxygen
penetration depth and the methane production, such as carbon content of the
sediments.

Directly after oxygen was removed from the incubation bottles and sediments from 5
– 15 cm depth were added, methane started to build up (Fig. 2). Algal biomass was
directly available for methane production, or the fresh organic matter enabled the
production of methane from previously present organic compounds (priming) or $CO_2$.
Methanotrophy may have been electron acceptor limited under the anoxic conditions,



and could not consume all methane produced. Even though the sediments recovered
directly after the establishment of anoxic conditions, and emitted methane, oxygen
pulse additions did decrease the methane release from the algal inputs.
A similar effect of an oxic-anoxic switch was observed by (Frenzel, Thebrath, and
Conrad 1990), who observed an abrupt increase in sedimentary methane emissions
when the oxygen concentration in the water overlying their core experiments dropped
below 18 µM. They assigned the difference between oxic and anoxic methane
emissions solely to an increased activity of methanotrophs under oxic bottom water
conditions.
Stable isotope analysis of the headspace methane in the stable, post-algal biomass
degradation phase (17 weeks of oligotrophic, and 11 weeks of eutrophic experiments,
Fig. 7), showed more negative $\delta^{13}CH_4$ values in the algal biomass experiments. The
$\delta^{13}C$ signal of the algal biomass likely decreased the $\delta^{13}CH_4$ values in the algal
addition experiments, with a larger effect in the oligotrophic lake, where the relative
contribution of algal biomass was largest, compared to the organic matter already
present in the sediments. Another potential factor is the shift in methanogenesis
pathway due to the algal biomass availability. (Zhou et al. 2022) showed that
cyanobacteria accumulation in lake sediments shifted the availability of organic
compounds for methane production and increased the potential for methylotrophic
methane production. Methylotrophic methanogenesis results in more depleted $\delta^{13}CH_4$
values compared to hydrogenotrophic methanogenesis (Summons, Franzmann, and
Nichols 1998). When comparing the oxic and anoxic experiments, only the oligotrophic
experiment showed significant differences: the $\delta^{13}CH_4$ values were lower (more



depleted) in the anoxic than in the oxic incubations, both with and without algal
biomass additions. This could also be caused by differences in methanogenesis
pathways, as hydrogenotrophic methane production (from $CO_2$) yields more $^{13}C$-
depleted methane than acetoclastic methanogenesis (Conrad 2005). As no changes
in the methanogenic community were observed between the oxic and anoxic
oligotrophic treatments, it is unlikely that a change in the community caused the
dominant methanogenesis pathway to swap and to cause the differences in the $\delta$
$^{13}CH_4$ values. A further explanation is that differences in rates of methanotrophy
caused the observed changes in $^{13}C$-compositions of methane. Indeed, increased
rates of methanotrophy under oxic conditions would be expected to contribute to a
less depleted isotopic composition of the remaining methane (Barker and Fritz 1981).

*Methane emissions and implications*
Sedimentation of (algal) biomass is a key factor in the magnitude and seasonal
variation in lake methane emission rates(Gruca-Rokosz and Cieśla 2021). Our
experiments with intact sediment cores, rather than slurries, showed a significant
decrease in methane emissions under oxic bottom water conditions compared to
anoxic bottom waters, similar to our slurry experiments. Algal biomass led to a strong
increase in methane emissions, which was dampened by oxygen exposure. Oxygen
was not actively mixed into the sediments: only the overlying water and headspace
were made oxic, oxygen penetration into the sediments was due to natural occurring
diffusion. Algal biomass was deposited on top of the sediments, and not mixed in
either, to mimic natural algal deposition. Both in the oxic and anoxic algal-addition
experiments, methane emission started almost immediately after algal biomass
addition (Fig. 3; Fig. S4). The weekly methane release was however lower under oxic



conditions and resulted in lower concentrations at the end of the 15-week experiment,
despite the methanogenic zone of the sediments (> 5 cm depth) not being in direct
contact with either the oxygen or algal biomass.

Generally, only the sediment surface is affected by the oxygen conditions in the bottom
water; deeper sediments are anoxic, due to the low diffusion coefficient through
sediments. (Maerki et al. 2009) investigated the oxygen, carbon and nitrogen
dynamics of lake sediments, and stated that short term (weeks to months) oxygen
exposure is insufficient to change the reactivity spectrum of eutrophic Lake Zug
sediments, that the exposure times are too short for that. Our whole core experiment
shows, however, that despite the fact that the methanogenic layer is deeper in the
sediments than the bottom water affected layer, the conditions in the bottom water are
still of key importance for the methane emissions from the sediments following the
deposition of algal material, for example after an algae bloom in the surface waters.
(Maerki et al. 2009) also state that over 95% of the anaerobic mineralization in Lake
Zug sediments was due to methanogenesis, and that methane oxidation was
responsible for over half of the oxygen consumption at the sediment surface. If a
similar situation is the case in our eutrophic Lake Baldegg, changes in the methane
cycling are likely to have substantial effects on the carbon and oxygen cycling in the
shallow sediments.

Our experiments show that the effects of a short (1-3 week) oxygen exposure can
last for several months, i.e. decreasing methane emissions without changing the
methane-related microbial community (Fig. 5, Fig. S6, S5, S10). We believe these
findings should be further explored in environmental settings. In certain Swiss lakes,



artificial aeration is already applied to combat bottom water anoxia. If brief pulses of
oxygen, like the 1- and 3-week oxygen exposure periods tested here, have the
capacity to reduce longer-term methane emissions, we believe this could be
promising, especially if applied directly after an algal bloom, as tested here. Given
the expectations of ongoing eutrophication in the upcoming decades, plus the global
warming of lakes that further draws down oxygen levels, we believe this should be a
topic for further research.

**Data availability statement**
Raw reads of the 16S rRNA sequencing data is deposited and made publicly
available in the online repository NCBI SRA, under accession number XXX (in
progress).

**Author contribution statement**
Conceptualization by SvG, MAL and CJS. Original draft preparation by SvG, review
and editing by SvG, NM, CG, MAL and CJS. Investigation and Methodology by SvG,
NM and CG.

**Acknowledgements**
The authors thank Patrick Kathriner, Karen Beck, Kathrin Baumann, Cameron
Callbeck and Dimitri Meier for help in the field and the lab.

**Competing interests**
At least one of the authors is a member of the editorial board of BG. The authors

have no further conflicts of interest to declare.



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
