# Peer review of "The effect of a short oxygen exposure period on algal biomass degradation and"

_EGUsphere, 2024_

## Author Comment (AC1)

The effect of a short oxygen exposure period on algal biomass degradation and methane release from eutrophic and oligotrophic lake sediments

Sigrid van Grinsven1* , Natsumi Maeda1 , Clemens Glombitza2 , Mark A. Lever2 4 , Carsten J. Schubert1,2

We want to thank the reviewer for the throughout review of this manuscript and the many valuable suggestions, both on style and content. We have replied to the specific comments below.

We copied the reviewers comments to a word file to be able to put our replies in line, for an easier overview of comments and replies. This has however messed up the layout of the reviewer's comments, and we are sorry for any inconvenience this causes.

General comments

Making a quick judgement, short term oxygen addition decreasing methane production from lake sediments for a longer period is an important finding. It is also self-evident, since oxygen disturbs methanogenesis and also increases methane oxidation.

A closer look makes things a bit more complicated: at shallow lakes water column has overturns and $CH_4$ may be oxidized in spring and autumn (Kankaala et al. 2007) and despite this can have large $CH_4$ emissions during open water period. Furthermore, methane oxidation in water columns may consume most of the produced methane in any case. In addition to this, aeration has other effects or not the effect wanted (Kuha et al. 2016; Niemistö et al. 2020). For a reader from northern Europe, it may be beneficial to explain how these studied lakes behave in general in order to be able to compare outcome of this study to previous ones. Now methods are lacking important things like water table depth, ice cover, possible overturns etc. On the other hand production of VFA:s don't connect to GHG balances, when no overall estimate of carbon additions and gas release is not done.

Thank you for these comments on describing the lake in more detail, which are shared by both reviewers. We will add more relevant details, and will in addition refer to data published elsewhere about these lakes. But we will also make the text on the lakes more explanatory.

We will also include additional calculations on carbon balance estimates.

It may be worth look and maybe cite studies of oxygenation (or aeration) in lake scale experiments (Pajala et al.2023) or generally effects to overall GHG balance of lakes (Huttunen et al. 2001). Notion that decrease in $CH_4$ may not be the only change in GHG:s following oxygenation may broaden the view on this topic. Maybe sentence "Over the course of the experiment, gas samples for $CH_4$, $CO_2$ and $N_2O$ analysis were (r.240)" hints to broader perspective on this related to GHG balance, however $N_2O$ is not mentioned later in this MS. Shall there be an other MS including $N_2O$ or can $N_2O$ be omitted here just by saying that " aeration had only minor effect to $N_2O$ production".

We will remove the part about N2O, because we focus in this manucript on carbon dynamics and don't want to confuse the reader. Thank you for pointing this out.

Oxygen penetration to sediment in natural settings is also different when water is oxygenated compared to experimnents in bottle. In this MS the problem is solved by taking upper 0 - 5 cm and 5 - 15 cm to slurry experiments, mimicking oxygen penetration depth by this way and also by doing core experiments.

Lot of work has been done (incubation VFAs, microbes), but there is lack of basic information. In the MS depth of the lakes sampled, sampling depth of sediment used in experiment is not expressed and also some other details are missing (listed below) making comparison di icult to earlier studies. So, it is also possible that these lakes are deeper, possible meromictic and having no overturns or ice cover, so results in these studies (above) may not be comparable to previously mentioned lake studies on boreal zone in more shallow lakes. Furthermore, using expression "build up to headspace in a week" makes comparisons to earlier studies even more complicated.

We will convert the results and figures to more general units, refering to methane emissions per volume of sediment, or sediment surface, to make sure that the readers can compare our results more easily to other published findings. Thank you for pointing this out.

Unfortunately I can not evaluate extensive microbial part of the MS due to my lack of knowledge. Did it bring something new to science – I can not say.

Overall, I think this is quite well written study and practically useful study about aeration in general. There is some important background information missing and also a fundamental mistake in $CH_4$ stable isotopes (is change from -70 per mill to -40 per mill depletion) and then in related discussion.

There is also lot of small typos. In my opinion, MS can be accepted after major revision addressing my comments and questions

Specific comments & technical corrections combined

r. 30 "in the eutrophic sediments" – is it rather "sediments from euthropic lake"
r. 41 and elsewhere in many places, space lacking: area(Bastviken et al. 2004); r.168 up to 6 cm(Cai

and Sayles 1996)

We are sorry that the formatting got mixes up. We will correct this is the next version of the manuscript and will in general comb through the manuscript in detail to remove any spelling and typing mistakes.

r. 144; Field sampling depth?

r: 160: The experimental setups were designed to mimic oxygen intrusion from overlying oxic 160 water. The experimental setups were designed to mimic oxygen intrusion from overlying oxic water. Something about how oxygenation/aeration is done in general in these lakes– by surface aerobic water pumping?

We will include text in the introduction and the methods, and where useful, in the discussion, about the conditions and situation in the lakes. We recognize that this will improve the manuscript and make it more interesting and easier to interpret for the reader.

r, 168 Let there be more space!: (marine) sediments up to 6 cm(Cai and Sayles 1996)

r. 187 "Slurry experiments were performed in triplicate in 0.5L Schott bottles, filled with a final volume of 258 ml sediment slurry". How big is the headspace then? According to our Schott "500 mL" bottles, their volume is ~607 mL, so was gas volume of headspace when sampling ~360 mL (607 mL – 258 mL + 10 mL). Is this the volume in calculation of µM in headspace?

We measured the volume of 5 of our Schott bottles (using water), averaged the resulting value, and used that for the headspace calculations.

r. 191 Is there any further data of added algae (C%, N%, d13C etc?) and is it possible to redo them? "0.1 g of freeze-dried Chlorella; DietFoods CH) was added to a subset of the bottles"

Yes, we have CN values of the algal biomass, and we will add these to the next version of the manuscript. These will also be used to bring a more detailed carbon balance.

r. 197 Was amount of this inoculum said somewhere. Inoculum comes a bit clearer on row. 301? ...oxygen, after which they received additional anoxic sediments, according to Table 2. "after anoxic conditions were established and the deeper anoxic sediments were added"

We tried to be concise and not repetitive, we therefore choose to refer to the table. We can think about a format that is more clear, though.

198 Maybe first the sediments in chapter... The resulting oxygen trends are shown in Fig. S2. Sediments were added via the

We don't understand this comment.

r. 203: Prior to gas extraction, 10 ml of N2 or air was pushed into the bottle. Immediately after, 10 ml of headspace gas was sampled and stored in N2 flushed 70 ml serum bottles.: Did you calculate how much of CH4 gas was lost due to dilution and sampling 10 mL (11 -15 times) during the experiment. Is this enough to lead to lag phase, when CH4 production decreases (row. 305)?

The dilution factor was the same at each sampling event. We don't understand exactly what this comment refers to, as row 305 does not mention any lag phase.

r. 191: (0.1 g of freeze-dried Chlorella; DietFoods CH); Why not to estimate amount of added carbon and d13C valuers of algae?

As mentioned before, we will do this in the next version of the manuscript. Thank you for the valuable suggestion!

r. 218 – 219 ... 25 cm of overlying water. To allow for headspace gas extraction over the experiment duration, 12 cm of overlying water was carefully removed without: why not to tell the headspace volume also (or only) ~942 mL? (pii*5 cm ^2*12 cm)

We will add this, thank you for the suggestion.

237 A word lacking: conditions in headspace (?) were indeed retained over the full course of the oxic experiment, as was

We will adapt this, thank you for the suggestion.

254 superscript here δ13C: ratios of methane 13C/12C (presented in the standard δ13C-notation relative to the

We will adapt this, thank you for the suggestion.

305 Did I calculate this right: 0.37 mg day, if diam 5 cm, then ~190 mg/CH4/m2/day? This is quite much and would be nice to know how much carbon was added. The methane concentration in the oligotrophic slurries plateaued (the emission almost 25-fold under permanently anoxic conditions (to 161 µM per week)

We will convert the values in the next version of the manuscript to more common units, to allow for easier interpretation.

315 Does this plateu mean that easily degradable carbon ends (CH4 +CO2 +WFA-C + DOC), or production is small and only compensating for $CH_4$ lost in samplings? incubations (Fig. 2). The methane concentration in the oligotrophic slurries plateaued and remained constant

It is likely that microbial methane oxidation is occuring in the experiments as well and is decreasing the net emission. We are not able to assess the contribution of methanogenesis and methanotrophy based on our methane measurements, but we do have microbial community data that shows an abundance of methanotrophs. Therefore, we expect that the plateau occurs because methane production and consumption are balanced at the timepoint the plateau appears. If methane production would completely stop, we would see a net decrease in headspace CH4. This is however not observed. It is possible that the production remains as high, but the consumption has increased strongly over time and therefore the net emission decreases. However, it is likely that the most labile compounds are indeed consumed at that timepoint, and that also contributes to the plateau.

317 and 326: Clewer way to express this but for data after use and comparisons it may be advisable to expresstheseasµmol/g,µmol/cm2orevenµmol/cm3. Andissemissionarightwordwhenitisbuild up to headspace and no area or tyime included ? 1900 (anoxic), 1500 (1-week oxic start) or 1200 (3- weeks oxic start) µmol per L headspace" Also in Sup. Tab 1: a. Concentration in headspace in µM after 28 weeks (constant since 14 weeks)

We will indeed change the units to more common units, to make it more comparable to literature.

r 302: All given concentrations and concentration increases are given in µM per liter headspace volume, not total amounts?

We do not understand what the reviewer refers to as total amounts, but we will convert the units to more common units to prevent this confusion.

497 Superscript 13 and WPDB was already in methods: given in the standard δ13C-notation, relative to the Vienna Pee Dee Belemnite (VPDB) reference.

We will adapt this, thank you for the suggestion.

r. 498 - 499 unnessessary space between rows: 13CH4 values in the algal biomass experiments in both the eutrophic and oligotrophic slurry

We will adapt this, thank you for the suggestion.

r. 505 Does this δ13C scale need to begin from zero? Fig. 7. CH4 values of headspace m

We will adapt this, thank you for the suggestion.

r. 525 Can there be a table of site characters already in methods, including this kind of data? Also C and N, BD etc. of sediment. Even average annual temperature and precipitations helps to get data on global scale. Now reader have no idea of lakes di erences except for size and phosphorus: historic and current carbon inputs, the TOC concentrations in sediments of both lakes

Yes, as mentioned above, we will add this. Thank you for the suggestion.

r. 545 Sampling depth on lake, sediment layer depth or sample depth in sediment profile? other studies indicate that the e ect of sediment depth is of stronger e ect(Yang et al.

We will adapt this to bring more clarity in the next version, thank you for the suggestion.

r 541& r. 602 space!: after which methane production rates stabilized (T. Wang et al. 2023). r. 588 Is it still "recent" in January 2025? A recent study by (Yang et al. 2021)
r. 612 added after oxygen removal. algae ?
r. 625 conditions ((Sobek et al. 2009; Huttunen et al. 2006).

We will adapt this to bring more clarity in the next version, thank you for the suggestions.

r.634 If you know pH, temperature, pressure and water amount and headspace volume this is possible withHenryslaw–atleastroughestimateindiscussion.
However,duetodi icultiesintranslating headspace CO2 concentrations to dissolved CO2,

We unfortunately do not have pH values in high enough resolution to make a confident estimate, and we therefore choose to not include such calculations.

r. 636 released as $CO_2$. Part of the produced $CO_2$ will again be converted prior to release to (to CH4, acetate?)

r. 643 As CO2 has a much lower global warming potential per mole than methane (a
r. 644 28 times lower on a hundred year basis, (Forster et al. 2021) ) the release of CO2 is converting

CH4 to CO2…
r. 647 Can you begin a sentence with parentheses? and is stored as such in the sediments. (Sobek et

al. 2009) published a weak linear 3

Thank you for pointing our these textual unclarities, we will correct them according to the suggestions made.

r. 670 For me it looks that algal additions increased d13C values (not decreased as written in MS) substantially (from ~-70 per mill even to -35 per mill). This changes also discussion that follows: Since O2 amendment had only slight e ect towards more positive values, this show that CH4 was not oxidized heavily. Is then only possible explanation then that e ect was due to Spirulina biomass having high d13C values. Can you still check d13C values for algae you used? If company made algae using carbonates (not air ebullition solely), also values towards quite positive d13C values for C are possible. Also sediment bulk d13C (C % + N%) as an background information would help in this? showed more negative 13CH4 values in the algal biomass experiments.

We indeed have d13C values of both algae types, we will include these in the next version of the manuscript and also spend more space in the discussion on the issue brought up here. We thank the reviewer for the information!

r. 675 Sentence begins with (). pathway due to the algal biomass availability. (Zhou et al. 2022) showed r. 696 Space! variation in lake methane emission rates(Gruca-Rokosz and Cieśla 2021). O

r. 705 Algae addition increases also oxygen consumption aerobically, which favors methanogenesis: experiments, methane emission started almost immediately after algal biomass

We will add this to the discussion.

r. 711 Generally, only the sediment surface is a ected by the oxygen conditions in

r, 728 Is this a beneficial result, if we think only CH4 and other GHG:s? Can it be that if oxygenation is done in right time of year so that settled algae is used already in aerobic processes before converting to $CH_4$ in anaerobic bottom, that this kind of method will be e ective in reducing $CH_4$ emissions withoutyearroundoxygenation? Ourexperimentsshowthatthee ectsofashort(1-3week)oxygen exposure can...

Yes, we think this could be an effective method, especially in lakes that have a relatively well defined algal bloom season/period, so it would be possible to pinpoint the right timing. Or after defining the need for temporary aeration based on (automated) Chlorophyll measurements. We however did not want to over-interpret our results, and therefore did not include such an outlook to the manuscript. We could, however, add such a paragraph to the manuscript if the reviewer considers this beneficial.

About figures: why not to put in same figure also CO2 (now in SUP). There reader sees immediately that anaerobic degradation is not producing equal amounts of CH4 and CO2 as methanogenesis theory suggests.

We choose to present the figures like this to not overload the reader with data, and rather focus on methane throughout the manuscript. We will again try out and see whether we can create attractive figures combining CH4 and CO2.

Worth reading and maybe citing:
Negligible e ect of hypolimnetic oxygenation on the trophic state of Lake Jyväsjärvi, Finland, Jonna K.

Kuhaa,∗, Arja H. Palomäki b, J. Tapio Keskinena,c, Juha S. Karjalainena Limnologica 58, 2016.

Huttunen JT, Hammar T, Alm J, et al (2001) Greenhouse Gases in Non-Oxygenated and Artificially Oxygenated Eutrophied Lakes during Winter Stratification. J Environ Qual 30:387–394. https://doi.org/10.2134/jeq2001.302387x

Kankaala P, Taipale S, Nykänen H, Jones RI (2007) Oxidation, e   lux, and isotopic fractionation of methane during autumnal turnover in a polyhumic, boreal lake. J Geophys Res Biogeosciences 112:1– 7. https://doi.org/10.1029/2006JG000336

Kuha JK, Palomäki AH, Keskinen JT, Karjalainen JS (2016) Negligible e   ect of hypolimnetic oxygenation on the trophic state of Lake Jyväsjärvi, Finland. Limnologica 58:1–6. https://doi.org/10.1016/j.limno.2016.02.001

Niemistö J, Silvonen S, Horppila J (2020) E   ects of hypolimnetic aeration on the quantity and quality of settling material in a eutrophied dimictic lake. Hydrobiologia 847:4525–4537. https://doi.org/10.1007/s10750-019-04160-6

Pajala G, Sawakuchi HO, Rudberg D, et al (2023) The E   ects of Water Column Dissolved Oxygen Concentrations on Lake Methane Emissions—Results From a Whole-Lake Oxygenation Experiment. J Geophys Res Biogeosciences 128:1–16. https://doi.org/10.1029/2022JG007185

Thank you, we will look into these!

---

## Author Comment (AC2)

**General comments**

The preprint numbered egusphere-2024-3979 investigated methane emissions from sediments from both eutrophic and oligotrophic lakes and the effect of additional algal organic carbon inputs, and examined the corresponding microbial community, which falls right into the scientific scope of BG. However, the similar studies have been published in recent years, and no substantial new finding was reported in the preprint. In addition, the manuscript was not organized in a good state. Specifically, the results were not supported by statistical analyses, the units of parameters are not standard, and the figures are not as meaningful as they should have, many sentences are not supported by literatures, and some sentences and paragraphs are redundant.

We thank the reviewer for the throughout reviewing of our manuscript and the very helpful comments. We have added specific replies to the individual points raised below.

**Specific comments**

**ABSTRACT**

It is argued that 3-week pulse of oxygen lowered the emitted methane for several weeks after anoxic conditions were re-established. However, this is because the 3-week pulse of oxygen decreased methane emission, which cause the total amount of methane in the headspace is lower than the control, not because oxygen pulse decreased methanogenesis rater after oxygen removal. It is suggested to re-evaluate the methanogenesis rate in different stages of the incubation based on the oxygen conditions.

We tried to convey the point that the measured methane emission rate per day in anoxic sediments is lower when those same sediments were previously exposed to oxygen. We however understand that this point is not conveyed well, and we will change the manuscript to bring this across more clearly, in both the text and the figures.

**INTRODUCTION**

It is well known that there are little algae in the oligotrophic lakes, so the influence of algal organic matter on methanogenesis in oligotrophic lake is seldom concerned. Why did you concern the influence of algal organic matter input on methanogenesis in oligotrophic lake? No sufficient reason was explained.

The mentioned lake has experienced nutrient-rich phases in the past (published elsewhere). We will make sure to refer to this in future versions of the introduction. Another reason is that eutrophication is an ongoing problem in many lakes worldwide.

The introduction is not well supported by references, many references should by supplemented in L41-49,L56-60, L66-75, 86-92, 100-106, 108-119.

The main hypothesis of the study should be explained and highlighted.

We thank the reviewer for raising these points and will adjust future versions of the manuscript to include this.

**METHODS**

More information about the two lakes should be supplemented, such as water depth, Chl a in water, hypoxia situations, etc.

Basic physical and chemical sediment properties should be supplemented.

We have not provided these here to keep the manuscript concise, but we understand now that it is good to mention more background on the lakes, both reviewers have suggested this. We will report them and also refer to publications where information on these lakes has been published previously.

L150, why 10 ℃?

We choose this temperature because it was close to the temperature in the lake itself, and we were able to use a temperature room at this temperature.

Table1, why there was no N2 flushed after 1 week treatment in Lake Baldegg?

We choose to not persue to 1 week treatment in Lake Baldegg because our results of Lake Lucerne had shown little difference between the 1 week and 3 week treatments.

L193, how to examine oxygen by Pyroscience? How to avoid the influence of slurry?

We used oxygen sensor spots that are glued inside the bottle, but that can be read out from the outside of the bottle. We will clarify this in the next version of the manuscript.

L215, why whole core experiment was not performed in the eutrophic lake?

We used our results from the Lake Lucerne study to decide on experiments in Lake Baldegg. Our results showed that whole core experiments showed large variations

and therefore were not the best way to perform experiments, therefore we choose to go for the bottle setup for the eutrophic lake.

L224, why Chlorella and Spirulina were chosen in the study? Are they the predominant in the studied two lakes?

This was mainly a practical choice, as these could be obtained. We will add more information on the used algal mass in the next version of this manuscript, as we have data available on the N and C content and isotopic values of both algae.

L227-230, how did you deposit algal biomass on the sediment surface, as freeze-dried algal cells usually are lighter than water. In addition, will the flushing of N2 influence the distribution of algal biomass?

Our algae-solution remained on the sediments, after careful deposition with a pipette. The N2 fliushing did indeed stir up the algal biomass, but after the flushing was ended, the material sunk down (the color of the overlying water was monitored to check whether the algal biomass was indeed on the sediment or present in the water). No algal biomass was observed at the water-headspace interface, showing that the algal biomass did not float.

L241-242, 10ml gas was sampled, why 10-15 ml gas was added?

Enough gas was added for a slight overpressure, to prevent the intrusion of oxygen.

**RESULTS**

Methane concentration in the headspace is the direct result from the examination, it is not suitable use the change of methane concentration to reflect the methanogenesis rate, such as Fig.1. The rate of methane emission in slurry incubation should be attributed to unit time and unit mass (or volume) of sediment, and methane emission from core should attributed to unit time and unit area of sediment, thus readers are able to compare your results with previous results.

Thank you for this suggestion, we will change the graphs and results accordingly.

Many units are not standard, such as 9 μmol per week in L299, μM in L305, μmol per L in L317, etc. Week is seldom used in the calculation of methanogens rate, and day is suggested in this study. μM should be replaced by μmol L$^{-1}$.

μM stands for μmol L$^{-1}$, but we will make sure to change this to make it more clear. We had chosen the per week rate because we could in that way report the differences between the phases of the experiment and between the different experiments. We do however see now that this makes it difficult to compare our

study to other literature, and will therefore change the units throughout the manuscript to make it easier to interpret the results.

Data of Fig1 are from Fig2, so their order should be changed in the manuscript.

Thank you for noticing, we will change this in the next version of the manuscript.

It is argued that 3-week pulse of oxygen lowered the emitted methane from both types of sediments by 50%. However, this is because the 3-week pulse of oxygen decreased methane emission, not because oxygen pulse decreased methanogenesis rater after oxygen removal. In figure 2, the similar methanogenesis rates are expected in different treatments after oxygen removal.  It is more suitable to analyze rates of methanogenesis (Fig 2) based on the oxygen conditions in incubation (Table 1).

We purposely do not claim that methanogenesis is decreased, as we can indeed only draw conclusions on the net methane emission, not on the separate processes of methane production and methane consumption. We analyzed the microbial community, the reaction intermediates, and the isotopic values to be able to make an assessment of the processes occuring in the sediments.

Statistical analyses are needed in the results analyses, such as Fig 1, 6, 7, etc.

We will add error bars in Fig. 1. We are not sure what the reviewer is referring to in Fig 6 and 7, as these include error bars. Is this about significant differences between treatments? We will look into methods to analyse this.

The community structure of methanogens is quite important in the study, they should be put in the text and with special concerns.

Thank you, we will give this more emphasis and give it a clearer place in the text.

DISCUSSION

L529-541 are more suitable for the next section.

Thank you for noticing, we will change this in the next version of the manuscript.

L593-594, ?

We do not understand this comment.

L612-614, why, how did you know?

We make the assumption that methanogens are present predominantly in the sediments deeper than 5 cm, which are not exposed to oxygen at all in our

experimental setups. However, we will note more clearly in the next version of this manuscript that we make this assumption.

L633, the FIGS9, S10should appeared in the results firstly.

Thank you for noticing, we will change this in the next version of the manuscript.

On the base of the above suggestion, refine the main findings and polish the figures.

Thank you, we will do so.

**Technical corrections**

Reference styles in the text must be uniformed according to the requirement of BG.

L51, sedimentary methane production→methane production in sediment

L80, H2?

L108, Artificial aeration is not only applied in Switzerland.

L130, showing and shown are repeated.

L144-147, sampling in the two lakes were carried out in different months, how about the environmental parameters in the water and sediment surface? How many cores were sampled?

L240, N2O?

Fig1, the symbols are too similar between 1 and 3 week oxic.

Fig4, which one is figure A?

L537, what does OM mean?

Some parts of the manuscript are redundant, not limited to the following sections, L302-303, 365-366, 473-475, 493-498, 668-669, etc. So, refine the language in the revision

Thank you for these detailed suggestions, they strongly help us to improve the next verion of the manuscript.